# Machine Learning Based Assessment of Inguinal Lymph Node Metastasis in Patients with Squamous Cell Carcinoma of the Vulva

**DOI:** 10.3390/jcm14103510

**Published:** 2025-05-17

**Authors:** Gilbert Georg Klamminger, Meletios P. Nigdelis, Annick Bitterlich, Bashar Haj Hamoud, Erich-Franz Solomayer, Annette Hasenburg, Mathias Wagner

**Affiliations:** 1Department of General and Special Pathology, Saarland University (USAAR) and Saarland University Medical Center (UKS), 66424 Homburg, Germany; 2Department of Obstetrics and Gynecology, University Medical Center of the Johannes Gutenberg University Mainz, 55131 Mainz, Germany; 3Department of Gynecology, Obstetrics and Reproductive Medicine, Saarland University Medical Center (UKS), 66424 Homburg, Germany

**Keywords:** vulvar cancer, lymph node metastasis, machine learning, artificial intelligence, cancer

## Abstract

**Background/Objectives**: Despite great efforts from both clinical and pathological sides to address the extent of metastatic inguinal lymph node involvement in patients with vulvar cancer, current research attempts are still mostly aimed at identifying new imaging parameters or superior tissue diagnostic workflows rather than alternative ways of statistical data analysis. In the present study, we therefore establish a supervised machine learning algorithm to predict groin metastasis in patients with squamous cell carcinoma of the vulva (VSCC) based on classical histomorphological features. **Methods**: In total, 157 patients with VSCC were included in this retrospective study. After initial exploration of valuable clinicopathological predictor variables by means of Spearman correlation, a decision tree was trained and internally validated (5-fold cross-validation) using a training data set (n = 126) and afterwards externally validated employing a holdout validation data set (n = 31) using standard metrices such sensitivity, positive predictive value, and AUROC curve. **Results**: Our established classifier can predict inguinal lymph node status with an internal accuracy of 79.4% (AUROC value = 0.64). Reaching similar performances and an overall accuracy of 83.9% on an unknown data input (external validation set), our classifier demonstrates robustness. **Conclusions**: The presented results suggest that machine learning can predict groin lymph node status in VSCC based on histological findings of the primary tumor. Such research attempts may be useful in the future for an additional assessment of inguinal lymph nodes, aiming to maximize oncological safety when targeting the most accurate diagnosis of lymph node involvement.

## 1. Introduction

Several risk factors, such as depth of invasion and lymphovascular space invasion (LVSI), have already been identified in the rare gyneco-oncological disease of squamous cell carcinoma of the vulva (VSCC), which is the most common tumor entity of all vulvar cancers (VC). However, from a histopathological point of view, other factors such as ulceration and infiltration into small veins remain of uncertain potential [1,2,3], not to mention emerging histological biomarkers such as tumor budding and stromal tumor infiltrating lymphocytes (sTILs), which have already proven their prognostic value in various tumor entities, but current evidence in VC remains scarce due to the small number of studies performed [4,5,6]. However, there is no doubt that in VSCC, the histologically confirmed tumor involvement of inguinal lymph nodes is the strongest prognostic factor not only in terms of survival but also for distant recurrence [7,8,9,10,11]. Consequently, in routine oncologic practice, the greatest possible effort is made both clinically (ultrasound imaging of inguinofemoral lymph nodes, optional fine needle aspiration or core needle biopsy, specific mandatory requirements for sentinel lymph node (SLN) sampling) and pathologically (optional intraoperative frozen sections, serial lymph node sections, so-called cytokeratin-based ultrastaging) to detect the presence of neoplastic cells within the lymphatic drainage pathways [12,13,14].

Employment of machine learning algorithms (a subfield of artificial intelligence focusing on the implementation of mathematical algorithms for data analysis) has demonstrated the possibility of predicting lymph node status in various tumor entities, such as cervical cancer and endometrial cancer, but also non-gynecological malignancies [15,16,17,18,19,20,21,22,23,24]. To our knowledge, similar research approaches in VC have not yet been reported. In the present study, we therefore analyze clinicopathological parameters of 157 patients with VSCC using a supervised machine learning approach in order to predict the absence/presence of neoplastic cells within inguinal lymph nodes and classify between patients with tumorous groin lymph node involvement and patients without inguinal metastatic spread.

## 2. Materials and Methods

Within this study, 157 patients who were diagnosed at the Institute of Pathology at Saarland University between January 2007 and December 2023 and were identifiable via the internal clinic software were included for subsequent case evaluation and analysis. Inclusion criteria from a clinical side were: (1) histomorphologically diagnosed squamous cell carcinoma of the vulva; (2) the patient underwent vulvectomy/wide excision, and tumor margins were subject to objective light microscopical assessment. Lymph node status was assessed in all tumors either according to current European guidelines or, years ago, via more invasive methods. For tumors with ≤1 mm depth of invasion, clinical groin evaluation alone was deemed sufficient in case of suspected negative lymph node affection. Sentinel lymph node (SLN) biopsy was performed for tumors <4 cm without clinical or radiographic evidence of inguinal lymph node involvement or multifocality. Inguinofemoral lymphadenectomy was conducted for tumors >4 cm or multifocal lesions. Bilateral groin treatment was mandatory for non-lateralized tumors. As exclusion criteria, solely high-grade squamous intraepithelial (HSIL) lesions as well as vulvar tumor entities other than squamous cell carcinomas (melanomas, adenocarcinomas) and recurrent lesions were a priori defined. This study was approved by the Ethics Committee of Saarland (study identification number 249/23); data were handled in alignment with the Declaration of Helsinki [25]. The histopathological workup of all surgical specimens was conducted in line with current diagnostic standards and national guidelines as described elsewhere [13,14]. Histomorphological tumor case/slide re-evaluation was performed on a multi-head light microscope (GGK, MN), TNM stages (including also perineural infiltration (Pn), vascular spread/infiltration (V), lymphovascular infiltration (L), and resection margin (R status) affections) were staged according to the actual TNM guidelines, 8th edition 2018 [26]. Additionally, depth of tumor infiltration (defined as the distance between the highest adjacent dermal papilla and the deepest infiltration of tumor cells) as well as the histological tumor grade (conventionally defined as tumor cell morphology in relation to the cell morphology of the tissue of origin) were noted. Within this study, all positive N stages were defined as “positive lymph node affection”, including the stage of so-called micrometastasis (neoplastic affection ≤2 mm; n = 2).

Initially, a Spearman Rho analysis was performed to explore correlations within the non-binary variables of our data set; the threshold for statistical significance was set at *p* < 0.05 (approximate *p* value for nonparametric correlation). Prior to the following supervised machine learning analysis, the data were exploratorily visualized using non-linear t-distributed Stochastic Neighbor Embedding (t-SNE). To avoid overfitting when training a classification algorithm, the data were split into a training cohort as well as an external validation cohort, employing a traditional 8:2 (training set:test set) data split [18]. The MATLAB Toolbox (MathWork, Natick, MA, USA; MATLAB Version R2024a) for Statistics and Machine-Learning™ and the MATLAB Classification Learner App were used to perform classifier training as well as model selection. Optional optimization of hyperparameters was waived. As a response variable, groin lymph node involvement (positive/negative; see Appendix A) was set and, besides the classical histomorphological parameters (vascular as well as lymphovascular and perineural infiltration/spread, tumorous affection of resection margins), clinicopathological variables positively correlated with groin lymph node metastasis in the previous correlation analysis served as predictor variables (tumor stage, infiltration depth, histological tumor grade).

Model validation was performed using a two-step process: After the classifier performance of the training data set was reported based on an internal 5-fold cross-validation, the trained model was validated using the holdout external validation cohort. Standard metrics (sensitivity, specificity, positive predictive value, negative predictive value, AUROC curve) were employed for performance evaluation, and a chi-square feature ranking algorithm facilitated the identification of the model’s feature importance.

## 3. Results

In total, 157 patients with histomorphologically diagnosed squamous cell carcinoma of the vulva matched our defined inclusion criteria and were included in this study. A summary of the clinical characteristics of our study cohort is depicted in Table 1. Hereby, 30 (19.1%) patients were diagnosed with a T1a tumor stage, 108 (68.8%) patients with a T1b tumor, and 19 (12.1%) patients with a T2 tumor. Overall, 33 (21.0%) patients had metastasis to groin lymph nodes (N positive, see Appendix A for a detailed depiction of the extent of lymph node involvement within the presented cohort). 25 (15.9%) tumors were reported as HPV-associated, and 54 (34.4%) tumors were reported as HPV-independent. Hereby, the viral etiopathology was determined during the diagnostic process either by means of molecular testing (in situ hybridization) or p16 immunohistochemistry. Overall, 78 (49.7%) tumors were without available p16/molecular testing since in these cases, viral testing was not performed for routine diagnostic reasons (in such cases, the time of initial diagnosis dates back several years). They were classified in alignment with the current *2020 WHO Classification of Female Genital Tumors* as Squamous cell carcinoma NOS (Not Otherwise Specified) of the vulva; see Appendix A.

An initial correlation analysis highlighted the association of most of the abovementioned evaluated clinicopathological parameters with groin lymph node involvement, as displayed in Appendix A. Since the parameter age was not correlated with the risk of lymph node metastasis, it was not further considered as a predictor variable in the subsequent machine learning analysis.

Employing a t-SNE dimension reduction prior to algorithm training, a distinct spatial separation of nodal positive tumors vs. nodal negative tumors was not feasible (Appendix A). In order to ensure robust algorithm performance, data were split into two separate cohorts: a training cohort for initial classifier training and an independent validation cohort for consecutive performance testing on previously unknown data. Our training data set included 126 patients (27 with groin lymph node affections and 99 without inguinal lymph node metastasis); our test data set comprised 31 patients (6 with groin lymph node affections and 25 without inguinal lymph node metastasis). As a supervised learning approach, we trained a tree algorithm to predict the class “absence of inguinal lymph node metastasis/N0” and achieved an internal accuracy (right classifications/all classifications) of 79.4%, with a corresponding AUROC value = 0.64 (Figure 1, positive class: “absence of inguinal lymph node metastasis/N0”). The TPR (true positive rate) for the detection of absent lymph node affections was 85.9% and correspondingly 55.6% for the correct classification of positive groin lymph node involvement. The resulting PPV (positive predictive value) for diagnosing a patient correctly as tumor stage N0 was 87.6%, and for identifying a lymphatic spread accurately employing our proposed algorithm and the histomorphological biomarkers put to the test, it was 51.7%. Table 2 depicts the classifier performance and metrics based on the internal classifier validation process.

A consecutively performed chi-square feature ranking algorithm allows further insight into the classification importance of our employed predictors, rating lympho-vascular space invasion, infiltration depth, and perineural infiltration as the most important features. See Appendix A for all ranked features and their respective scores. Validating the established algorithm on the holdout test set, sufficient classifier performance with an overall accuracy of 83.9% could be determined. The TPR of 96.0% and the PPV of 85.7% for predicting N0 tumor status, as well as the PPV of 66.7% for predicting groin lymph node involvement, indicate a modest yet robust classification performance. Results of the external validation process are displayed in Figure 2; for a tabular listing, see Appendix A.

## 4. Discussion

This is a retrospective study evaluating the potential of traditional histomorphological risk factors serving as predictor variables for a machine learning-based prediction of inguinal lymph node status in VSCC. As a key observation of our pilot study, the data presented indeed indicate that machine learning-assisted classification may be useful to predict the status of groin lymph nodes in patients with invasive vulvar tumors, with a modest overall accuracy of 83.9% on an external validation data set. Such approaches may be useful in supporting the extensive clinical and pathological efforts involved in both the preoperative clinical examination of the groin and the postoperative histological processing of surgical specimens. Although machine learning-assisted classification models have proven advantageous in the prediction of several oncological diagnoses and prognostic outcome parameters in gynecological cancers, most studies evaluate the potential of automated learning systems in cervical cancer as well as ovarian or endometrial cancer [27,28,29].

In contrast, within VC, solely individual applications have been reported; this research gap was especially highlighted in a recent review by Gandotra et al. [29]. In an emergent analysis, the team of Zhou et al. trained several data-driven classifiers (k-nearest neighbor [30], random forest, adaptive boosting, and latent Dirichlet allocation) using clinical data and pelvic magnetic resonance imaging (MRI) images of 52 VC patients to predict inguinal lymph node metastasis. Splitting the patient cohort into a training data set (n = 37), as well as a validation data set (n = 15), an integrated approach making use of all aforementioned models achieves an AUC of 0.717 (validation set) differing between patients with lymph node metastasis and patients without tumorous groin affections [30]. In a similar approach, the team of Garganese et al. established an ultrasound-guided machine learning model to classify groin lymph node metastasis based on 127 patients and previously reported distinct sonographic features by integrating different classification approaches (random forest classifier, regression binomial model, decision tree, similarity profiling) [31]. Their so-called “Morphonode Predictive Model” was made available as open source and provides different output variables, among others, a random forest-derived malignancy prediction, which yields an accuracy of 93.3% [31,32]. Another approach employed a random forest-based analysis on genomic data for the identification of disease-related genes [33]. To our knowledge, our study presented is the first aiming at the prediction of lymph node metastasis in VC using traditional histological parameters.

The primary strengths of our study are the focus on a cutting-edge topic as well as its response to a recently identified research gap [29]. In line with previous studies, we present the practical applicability of an easy-to-use machine learning-training environment utilizing the MATLAB Classification Learner App [34,35]. Using traditional histopathological features as predictor variables (which still represent the gold standard in diagnostics), our easily interpretable and pragmatic approach could be easily employed globally, even in low-resource settings.

As main weaknesses, one could consider our overall sample size as well as the lack of radiomic or genetic data integration. However, such a large number of predictor variables would require a higher number of tumor samples. Within the scope of our study, only a distinct number of specifically selected predictor variables were integrated, securing an appropriate alignment between the overall sample size and number of predictors (one in ten rule) [36]. Interestingly, our established machine learning algorithm uncommonly yields a superior overall accuracy on the external validation data set (83.9%) compared to the training data set (79.4%). This discrepancy is presumably due to the imbalance in size between the two data sets, as well as a potentially lower variability within the smaller test set, which may explain the observed variation in performance. That said, even though we identified most important features for our classification algorithm employing a chi-square feature ranking algorithm (Appendix A), it can be expected that individual decision characteristics will adapt/transform as the number of cases increases, leading to a deeper understanding of the most useful predictor parameters—in addition to a potential further optimization of overall performance. In that regard, our study cohort does not have enough power to test the ability to predict survival outcomes directly, which, on a side note, would be an interesting future study approach. Such efforts would ideally require definite knowledge about the most valuable predictor variables (histology, imaging, genetic) and would profit not only from a larger database but also from a multi-center approach by further minimizing the risk of data inherent bias. That said, our trained algorithm demonstrates superior performance in the “absence of inguinal lymph node metastasis/N0” class—this is likely attributable to a class imbalance, which may have caused the model to be biased toward the majority class. In future studies, potential strategies to address this underperformance in the minority class could include adjustments at the algorithmic level, such as class weighting. Finally, it is important to acknowledge another limitation of this study: all predictor variables were derived from histopathological examinations, which are available only postoperatively (VC surgery is usually performed together with a surgical treatment of the groin; see “*Materials and Methods*” above). However, within the scope of this proof-of-concept study, we aimed to establish and demonstrate the general feasibility of machine learning combined with histological features as a prognostic strategy in VSCC. Building upon these findings, future research can take a more practical approach, for example, by evaluating the prognostic relevance of histological markers obtained from primary diagnostic biopsies, possibly in combination with radiographic (e.g., ultrasound) features, in such contexts, even preoperative applications of the proposed method may be feasible.

Albeit several diagnostic approaches aim to detect lymph nodes involvement (e.g., ultrasound, histological ultrastaging), several clinical circumstances such as the optional bilateral lymphadenectomy in case of lateralized tumors >4 cm as well as the waiver of surgical lymph node assessment in pT1a (infiltration <1 mm) at all highlight the need for secure risk evaluation, id est detection of a (potential) groin lymph node metastatic spread [13]. Our results show that the prediction of the absence of inguinal lymph node metastasis in VSCC is feasible using solely histomorphological aspects of the primary tumor itself, while the classification of positive groin lymph node affection remains less accurate. Especially in a high-risk constellation (e.g., elderly patients with comorbidities and an increased risk of lymphedema or other complications), machine learning-assisted models may serve as markers for a more precise diagnosis, inter alia, potentially helping to compensate for a false positive/false negative SLN procedure [37,38]. That said, future machine learning approaches in VC could aim not only at the detection of inguinal lymph node involvement but also prediction of metastasis distribution by a number of positive nodes as important independent risk factors [10,39,40]. A different strategy may also employ an unsupervised learning approach aiming at the detection of yet unknown risk factors.

## 5. Conclusions

Our study demonstrates that a supervised machine learning algorithm may be capable of predicting the absence of inguinal lymph node metastasis in patients with VSCC based solely on histomorphological tumor aspects. The presented model achieved an internal accuracy of 79.4% (AUROC = 0.64) in predicting the status of inguinal lymph nodes and showed robustness with an external validation accuracy of 83.9%—that said, the prediction for positive groin lymph node status remains still less precise than the prediction of the absence of inguinal lymph node metastasis. Further research will determine whether the integration of histology and radiology may support or even surpass our conventional approaches to inguinal lymph node assessment.

## Figures and Tables

**Figure 1 jcm-14-03510-f001:**
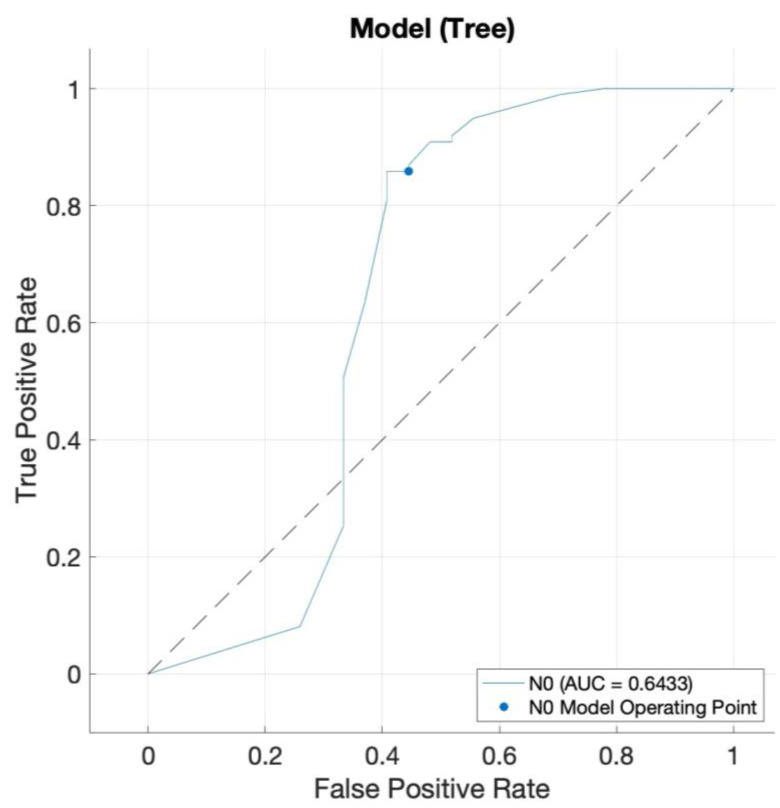
Receiver operating characteristic (ROC) curve of our decision tree classifier with an AUROC value = 0.64. The blue curve represents the classifier decision thresholds for predicting “N0” (id est no groin lymph node affection). The dotted black line represents the performance level of a random classification.

**Figure 2 jcm-14-03510-f002:**
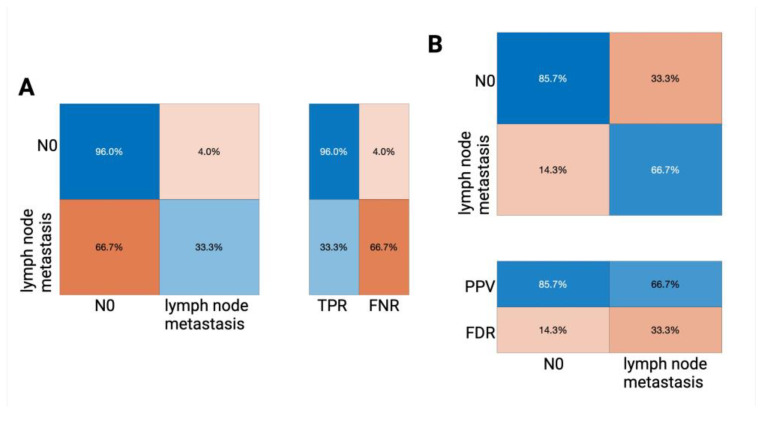
Confusion matrices illustrating the performance of the trained decision tree algorithm on our external validation (holdout) data set. The confusion matrix (a standard tabular representation in the field of machine learning that allows not only a detailed performance breakdown but also class-wise insight) visualizes the TPR (true positive rate) and FNR (false negative rate) (**A**) as well as the PPV (positive predictive value) and the FDR (false discovery rate) (**B**). In each matrix, the rows represent the actual class labels (i.e., the true lymph node status), and the columns represent the predicted class labels classified by the algorithm; the percentages indicate the proportion of cases within each category.

**Table 1 jcm-14-03510-t001:** Listing of clinical characteristics of our study cohort. L = invasion in lymphatic vessels (0 = no/1 = yes), N0 = negative lymph node affection, Pn = perineural infiltration (0 = no/1 = yes), R=resection margin (0 = tumor free/1 = tumor cells microscopically attached), T = tumor stage, V = invasion in blood vessels (0 = no/1 = yes).

Variables of Interest	N = 157
Age, years	66 (median, IQR: 53–79)
T1a	30 (19.1%)
T1b	108 (68.8%)
T2	19 (12.1%)
T3	-
N0	124 (79.0%)
positive groin lymph node affection (Nmic/N1a to N2c)	33 (21.0%)
L0	132 (84.1%)
L1	25 (15.9%)
V0	146 (93.0%)
V1	11 (7.0%)
Pn0	144 (91.7%)
Pn1	13 (8.3%)
infiltration depth (in cm)	0.7134 (mean), 0.8120 (std. deviation)
R0	129 (82.2%)
R1	28 (17.8%)

**Table 2 jcm-14-03510-t002:** Presents individual performance metrics and overall classification performance (overall accuracy and AUROC) of the established decision tree algorithm, based on internal validation using 5-fold cross-validation. The table is divided into two sections based on the response variable: prediction of absent (N0) versus positive groin lymph node involvement.

Key Performance Indicators of Our Tree Classifier Performance (Internal Classifier Validation)	Overall Accuracy = 79.4%
**no lymph node affection (N0):**	
TPR (true positive rate)	85.9%
FNR (false negative rate)	14.1%
PPV (positive predictive value)	87.6%
FDR (false discovery rate)	12.4%
**positive groin lymph node affection:**	
TPR (true positive rate)	55.6%
FNR (false negative rate)	44.4%
PPV (positive predictive value)	51.7%
FDR (false discovery rate)	48.3%
**AUROC value**	0.6433

## Data Availability

Please contact the corresponding author for individual solutions.

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
