# Peer review of "Machine Learning Based Assessment of Inguinal Lymph Node Metastasis in Patients with Squamous Cell Carcinoma of the Vulva"

_jcm, 2025, doi:10.3390/jcm14103510_

Round 1
Reviewer 1 Report
Comments and Suggestions for Authors・This paper examines whether a supervised machine learning approach can be used to predict the presence of inguinal lymph node metastasis in vulvar cancer.
・Classic histopathological findings and all categorical clinicopathological variables that correlate positively with inguinal lymph node metastasis in a correlation analysis are trained as predictors. This is an original study using a method different from conventional pathological findings of the primary tumor,
 sentinel lymph node biopsy, and various imaging techniques.
・Although the method is described, the actual technique is difficult to understand.
 It would be easier to understand if specific positive and negative images of lymph node metastasis were illustrated.
・Table 2 describes the relationship between positive lymph nodes and pathological prognostic factors.
 If the primary tumor is resected and no pathological evaluation is performed (e.g., radiotherapy), how is this predicted?
・Table 3 and Figure 2 are not clear. We believe that additional description of the contents is necessary.
・The negative lymph node group has a high accuracy rate, but the positive group has a low rate, which would be even lower if micro metastatic findings of 2 mm or less were not included.
 It is necessary to consider why such results are obtained.
・In practice, sentinel node biopsy is performed to determine whether dissection should be performed. Because of the high negative predictive value, this study may be useful in elderly patients with comorbidities and in helping to prevent lymphedema and other complications that may occur with lymph node  
 dissection.
・The references listed were adequate.
Author Response
Point-by-point response to reviewer 1
Manuscript ID: jcm-3529474
Title: Machine Learning Based Assessment of Inguinal Lymph Node Metastasis in Patients with Squamous Cell Carcinoma of the Vulva
This paper examines whether a supervised machine learning approach can be used to predict the presence of inguinal lymph node metastasis in vulvar cancer. Classic histopathological findings and all categorical clinicopathological variables that correlate positively with inguinal lymph node metastasis in a correlation analysis are trained as predictors. This is an original study using a method different from conventional pathological findings of the primary tumor,sentinel lymph node biopsy, and various imaging techniques. |
We thank the reviewer for all the work and thoughtful comments on our manuscript. In the following, we will present how we addressed the requirements of the reviewer and incorporated all points raised into our manuscript. |
Although the method is described, the actual technique is difficult to understand. |
We would like to thank the reviewer for pointing this out and have now illustrated specific positive and negative images of lymph node metastasis in our new Supp Figure 1. |
Table 2 describes the relationship between positive lymph nodes and pathological prognostic factors. If the primary tumor is resected and no pathological evaluation is performed (e.g., radiotherapy), how is this predicted? |
We thank the reviewer for bringing up this point, which is also of distinct importance for the interested reader; we therefore rephrased and clearly state now the required mode of lymph node assessment for study inclusion, see page 2 of the revised manuscript: “Lymph node status was assessed in all tumors either according to current European guidelines or, alternatively years ago, via more invasive methods. For tumors with ≤1 mm depth of invasion, clinical groin evaluation alone was deemed sufficient in case of suspected negative lymph node affection. Sentinel lymph node (SLN) biopsy was performed for tumors <4 cm without clinical or radiographic evidence of inguinal lymph node involvement or multifocality. Inguinofemoral lymphadenectomy was conducted for tumors >4 cm or multifocal lesions. Bilateral groin treatment was mandatory for non-lateralized tumors”.
That said, patients who underwent primary radiotherapy were not included in this study (inclusion criteria: patient underwent vulvectomy/wide excision) since no sufficient histopathological information is retrieved. |
Table 3 and Figure 2 are not clear. We believe that additional description of the contents is necessary. |
On a side note: Another reviewer suggested revising the table numbering. Accordingly, the former Table 3 - referred to in this comment - has been renumbered as Table 2 in the revised manuscript.
We appreciate the reviewer’s attention to this important point. In response to the request, we have revised and expanded the individual figure and table legends to ensure they are fully interpretable on their own, independent of the main manuscript text.
It states now for (former) Table 3 (page 5): “Table 2 presents individual performance metrics and overall classification performance (overall accuracy and AUROC) of the established decision tree algorithm, based on internal validation using 5-fold cross-validation. The table is divided into two sections based on the response varia-ble: prediction of absent (N0) versus positive groin lymph node involvement.”, and for Figure 2 (page 6): “Confusion matrices illustrating the performance of the trained decision tree algorithm on our external validation (holdout) dataset. The confusion matrix (a standard tabular representation in the field of machine learning that allows not only a detailed performance breakdown but also class-wise insight) visualizes the TPR (true positive rate) and FNR (false negative rate) (A) as well as the PPV (positive predictive value) and the FDR (false discovery rate) (B). In each matrix, the rows represent the actual class labels (i.e., the true lymph node status), and the columns represent the predicted class labels classified by the algorithm; the percentages indicate the proportion of cases within each category”. |
The negative lymph node group has a high accuracy rate, but the positive group has a low rate, which would be even lower if micro metastatic findings of 2 mm or less were not included. It is necessary to consider why such results are obtained. |
We are thankful for the reviewer’s attention to this important issue and added a distinct reasoning as well as possible future strategy within the limitation section of our discussion, see the revised version of our manuscript (page 7): “That said, our trained algorithm demonstrates superior performance in the "absence of inguinal lymph node metastasis / N0" class – this is likely attributable to a class imbalance, which may have caused the model to be biased towards the majority class. In future studies, potential strategies to address this underperformance in the minority class could include adjustments at the algorithmic level, such as class weighting”. |
In practice, sentinel node biopsy is performed to determine whether dissection should be performed. Because of the high negative predictive value, this study may be useful in elderly patients with comorbidities and in helping to prevent lymphedema and other complications that may occur with lymph node dissection. |
We appreciate the reviewer’s insightful observation regarding this point and added this information to our discussion, see page X: “…in such a high-risk constellation (e.g., elderly patients with comorbidities and an in-creased risk of lymphedema or other complications) machine learning-assisted models …”. |
The references listed were adequate. |
We thank the reviewer for the positive assessment. |
Reviewer 2 Report
Comments and Suggestions for Authors
Congratulations to the authors for choosing the interesting topic of machine learning-based assessment of inguinal nodal metastasis in patients with vulvar SCC. Machine learning based assessment is gaining more and more attention nowadays.
The article is well organized and structured.
Lines 88 – 89…please cite the current TNM guidelines.
For methods, please provide the date from when to when your study enrolled patients.
The main limitation of your study is that if you want to investigate inguinal lymph node metastasis in patients with squamous cell carcinoma of the vulva, you must completely remove the vulvar cancer surgically. This surgery almost always involves an inguinal SNB or lymphadenectomy.
All of the machine learning results you presented can be performed after surgery. If you have already removed the lymph nodes, the machine learning information is of less importance after the operation. You should discuss this in detail.
Author Response
Point-by-point response to reviewer 2
Manuscript ID: jcm-3529474
Title: Machine Learning Based Assessment of Inguinal Lymph Node Metastasis in Patients with Squamous Cell Carcinoma of the Vulva
Congratulations to the authors for choosing the interesting topic of machine learning-based assessment of inguinal nodal metastasis in patients with vulvar SCC. Machine learning based assessment is gaining more and more attention nowadays. |
We thank the reviewer for positive assessment of our work and also the thoughtful comments on our manuscript. Please refer to this point-by-point response as well as our revised version of the manuscript to see in detail how we addressed each point raised. |
The article is well organized and structured. |
Thank you very much. |
Lines 88 – 89…please cite the current TNM guidelines. |
As requested, we added the reference; see the new reference [26] within the revised version of the manuscript. |
For methods, please provide the date from when to when your study enrolled patients. |
We appreciate the reviewer for highlighting this important issue; it states no on page 2: “Within this study 157 patients who were diagnosed at the Institute of Pathology at Saarland University between 2007–2023 were included for subsequent case evaluation and analysis”. |
The main limitation of your study is that if you want to investigate inguinal lymph node metastasis in patients with squamous cell carcinoma of the vulva, you must completely remove the vulvar cancer surgically. This surgery almost always involves an inguinal SNB or lymphadenectomy. All of the machine learning results you presented can be performed after surgery. If you have already removed the lymph nodes, the machine learning information is of less importance after the operation. You should discuss this in detail. |
We are grateful to the reviewer for raising this significant point – we added this point to our discussion section on page 7/8 within the manuscript; it states now: “Finally, it is important to acknowledge another limitation of this study: all predictor variables were derived from histopathological examinations, which are available only postoperatively (VC surgery is usually performed together with a surgical treatment of the groin; see ‘Materials and Methods’ above). However, within the scope of this proof-of-concept study we aimed to establish and demonstrate the general feasibility of machine learning combined with histological features as prognostic strategy in VSCC. Building upon these findings, future research can take a more practical approach, for example by evaluating the prognostic relevance of histological markers obtained from primary diagnostic biopsies, possibly in combination with radiographic (e.g., ultrasound) features - in such context, even preoperative applications of the proposed method may be feasible”. |
Reviewer 3 Report
Comments and Suggestions for Authors
1.Table 1 needs to be redone. The classification of tumors needs to be seen more clearly.
- We suggest splitting the table 1 into several independent tables.
- Discussions should focus especially on the first part of Table 3, - no lymph node affection
- A lot of statistical data without much relevance. I suggest that you keep only the statistical data that is relevant to the objective of the study.
- The surgical value of the study and its implications for the management of future cases should be highlighted.
- The conclusion must be clearer and objectively the results of the study.
Author Response
Point-by-point response to reviewer 3
Manuscript ID: jcm-3529474
Title: Machine Learning Based Assessment of Inguinal Lymph Node Metastasis in Patients with Squamous Cell Carcinoma of the Vulva
We thank the reviewer for all the work and thoughtful comments on our manuscript. In the following, we will present how we addressed the requirements of the reviewer and incorporated the points raised into our manuscript.
Table 1 needs to be redone. The classification of tumors needs to be seen more clearly. We suggest splitting the table 1 into several independent tables. |
As requested by the reviewer, we modified Table 1 within the revised version of our manuscript (see page 4), setting a distinct focus on the clinical aspects / the classification of the tumors. We display the information regarding histological grading and HPV association of tumors now in the new Supp Table 2. Additionally, we adjusted the figure legend of Table 1 to be more clearly for the interested reader, see page 3. |
A lot of statistical data without much relevance. I suggest that you keep only the statistical data that is relevant to the objective of the study. |
We agree with the reviewer’s observation: machine learning analyses typically generate a substantial amount of data and statistical output. While we aim for conciseness and have therefore included a considerable portion of this information—such as the visualization of the t-SNE algorithm—in the Supplementary Material, we also wanted to provide interested readers with sufficient insight into our results.
In line with the reviewer’s request and to further enhance the clarity and presentation of our findings, we have moved the original Table 2 to the Supplementary Material. This information is now presented as the new Supplementary Table 3; please refer to the revised Supplementary Material. |
Discussions should focus especially on the first part of Table 3, - no lymph node affection. |
We are grateful to the reviewer for raising this significant point – especially since this topic was brought up also by Reviewer 1. Within our discussion, we now added a distinct interpretation of the performance of our classifier on the positive class (“N0”). Furthermore, we elaborate on possible future strategies to deal with data imbalance (page 7): “That said, our trained algorithm demonstrates superior performance in the "absence of inguinal lymph node metastasis / N0" class – this is likely attributable to a class imbalance, which may have caused the model to be biased towards the majority class. In future studies, potential strategies to address this underperformance in the minority class could include adjustments at the algorithmic level, such as class weighting”.
Additionally, we revised and rephrased further parts of our discussion / conclusion of our study (see e.g., line 328 and line 344/345 of the revised version of the manuscript), in line with the reviewers request. |
The surgical value of the study and its implications for the management of future cases should be highlighted. |
We appreciate the reviewer’s insightful observation regarding this point. Within the revised version of the manuscript, we now discuss the association of our proof-of-concept study on the one hand and the practical implementation of machine learning into the surgical workflow on the other hand. On page 7/8 within the manuscript; it states now: “Finally, it is important to acknowledge another limitation of this study: all predictor variables were derived from histopathological examinations, which are available only postoperatively (VC surgery is usually performed together with a surgical treatment of the groin; see ‘Materials and Methods’ above). However, within the scope of this proof-of-concept study we aimed to establish and demonstrate the general feasibility of machine learning combined with histological features as prognostic strategy in VSCC. Building upon these findings, future research can take a more practical approach, for example by evaluating the prognostic relevance of histological markers obtained from primary diagnostic biopsies, possibly in combination with radiographic (e.g., ultrasound) features - in such context, even preoperative applications of the proposed method may be feasible”. |
The conclusion must be clearer and objectively the results of the study. |
We very much acknowledge the reviewer’s valuable comment on this topicand adapted our conclusion accordingly, see page 8 of the revised manuscript: “Our study demonstrates that a supervised machine learning algorithm may be capable of predicting the absence of inguinal lymph node metastasis in patients with VSCC based solely on histomorphological tumor aspects. The presented model achieved an internal accuracy of 79.4% (AUROC=0.64) in predicting the status of inguinal lymph nodes and showed robustness with an external validation accuracy of 83.9% - that said, the prediction for positive groin lymph node status remains still less precise than the pre-diction of the absence of inguinal lymph node metastasis. Further research will determine whether the integration of histology and radiology combined may support or even surpass our conventional approaches of inguinal lymph node assessment.” |
Round 2
Reviewer 2 Report
Comments and Suggestions for Authors
The article has been improved taking into account all recommendations.
Please also add months….... between January? 2007 – December? 2023 in line 69.
Good luck with your further research.
Author Response
We sincerely thank the reviewer for the positive and encouraging evaluation of our manuscript. We greatly appreciate the time, expertise, and effort invested in the review process.
We corrected the line as requested.